# The Mutual Autoencoder: Controlling Information in Latent Code Representations

## Abstract

Variational autoencoders (VAE), (Kingma & Welling, 2013; Rezende et al., 2014), learn probabilistic latent variable models by optimizing a bound on the marginal likelihood of the observed data. Beyond providing a good density model a VAE model assigns to each data instance a latent code. In many applications, this latent code provides a useful high-level summary of the observation. However, the VAE may fail to learn a useful representation when the decoder family is very expressive. This is because maximum likelihood does not explicitly encourage useful representations and the latent variable is used only if it helps model the marginal distribution. This makes representation learning with VAEs unreliable. To address this issue, we propose a method for explicitly controlling the amount of information stored in the latent code. Our method can learn codes ranging from independent to nearly deterministic, while benefiting from decoder capacity. Thus, we decouple the choice of decoder capacity and the latent code dimensionality from the amount of information stored in the code.

## 1 Introduction

Latent variable models are a powerful approach to generative modeling of complicated distributions. In a latent variable model we model a distribution over observables $x \in \mathcal{X}$ through a hierarchical model,

$$p_\theta(x) = \int_{\mathcal{Z}} p_\theta(x|z) \, p(z) \, \mathrm{d}z. \tag{1}$$

In practice, we *learn* the model parameters $\theta$ in (1) using maximum likelihood estimation (MLE),

$$\max_{\theta \in \Theta} \mathbb{E}_x[\log p_\theta(x)]. \tag{2}$$

The recent variational autoencoder (VAE) method, (Kingma & Welling, 2013; Rezende et al., 2014), provides tractable lower bounds to (2) for deep latent models like (1).

The model (1) has another interesting property: although the latent variables $z$ are never observed, they provide a high-level summary of the observation $x$. Therefore $z$ could serve as a powerful representation in a number of machine learning tasks (Bengio et al., 2013). Because (1) can be learned from unlabeled data, we can in principle use latent variable models for unsupervised representation learning, an important building block in artificial intelligence systems.

But when does unsupervised learning lead to useful representations? We argue that trusting (2) alone is not enough to ensure that $z$ stores useful information.[1] In particular, the amount of information stored in $z$ depends on the expressiveness of the model $p_\theta(x|z)$ with respect to the true data distribution (Chen et al., 2017). In practice this has made the VAE approach difficult to work with in important applications such as natural language processing and for modeling discrete data.

We propose a solution in the form of explicit control of information flow in latent variable models. This is illustrated in Figure 1: optimizing $\log p_\theta(x)$ as in VAEs will lead to a model $\hat{p}$ which achieves a certain mutual information between $x$ and $z$ but this amount of information is difficult to predict and in fact may be zero (Chen et al., 2017; Zhao et al., 2017). Our *mutual autoencoder*

---

[1]See also the recent article (Huszár, 2017).

(MAE) approach forces information flow by ensuring that the estimated model $\hat{p}_m$ achieves a user-specified mutual information $M = I_{\hat{p}_m}$. Therefore, we precisely control the amount of bits stored in the representation but leave the organization and use of this information to be learned. To control information in this way requires novel algorithms and the rest of the paper discusses our procedure in more detail.

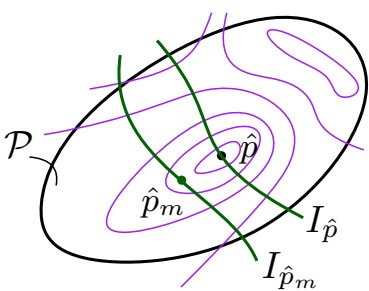

Figure 1: The mutual autoencoder maximizes the likelihood $\log p_\theta(x)$ while constraining model family $\mathcal{P}$ by ensuring that the latent variable $z$ and the observables $x$ share a desired mutual information $M = I_{\hat{p}_m}$. The purple lines denote level sets of the likelihood $\mathbb{E}_x[\log p_\theta(x)]$.

## 2 BACKGROUND: VARIATIONAL AUTOENCODERS

Consider a latent variable model with a data variable $x \in \mathcal{X}$ and a latent variable $z \in \mathcal{Z}$, $p_\theta(z, x) = p(z)\,p_\theta(x|z)$. Given the data $x_1, \ldots, x_n$, we would like to train the model by maximizing the marginal log-likelihood,

$$\mathcal{L} := \mathop{\mathbb{E}}_{x \sim q}[\log p_\theta(x)] = \mathop{\mathbb{E}}_{x \sim q}\left[\log \int_\mathcal{Z} p_\theta(x|z)\,p(z)\,\mathrm{d}z\right], \tag{3}$$

where $q(x)$ denotes the empirical distribution of $x$, $q(x) = \frac{1}{n}\sum_{i=1}^n \delta_{x_i}(x)$. However, the integral in (3) is intractable in many applications of interest. The idea behind variational methods is to instead maximize a lower bound $L(p_\theta, q)$ to the log-likelihood, where

$$L(p, q) := \mathop{\mathbb{E}}_{x \sim q}\left[\mathop{\mathbb{E}}_{z \sim q(z|x)}[\log p(x|z)] - D_{\mathrm{KL}}(q(z|x)\|p(z))\right]. \tag{ELBO}$$

Any choice of $q(z|x)$ gives a valid lower bound $L(p, q) \leq \mathcal{L}$ with a gap of $\mathbb{E}_{x \sim q}[D_{\mathrm{KL}}(q(z|x)\|p(z|x))]$.

VAEs (Kingma & Welling, 2013; Rezende et al., 2014) replace the per-instance posteriors $q(z|x)$ by an inference network $q_\theta(z|x)$ that is trained together with $p_\theta(x|z)$ to jointly maximize $L(p_\theta, q_\theta)$. For the inference network $q_\theta(z|x)$, this is equivalent to minimizing $D_{\mathrm{KL}}(q_\theta(z|x)\|p_\theta(z|x))$, so one can think of $q_\theta(z|x)$ as an approximate posterior for $p(z|x)$.

Being a stochastic mapping of data $x$ to a latent code $z$, the inference network $q_\theta(z|x)$ is sometimes called the *encoder* and, by the same analogy, the generator network $p_\theta(x|z)$ is called the *decoder*.

## 3 VAE OBJECTIVE IS INSUFFICIENT FOR REPRESENTATION LEARNING

A major appeal of the VAE framework is the ability to learn meaningful latent codes $z$ from unlabeled or only weakly labeled data. Despite numerous promising results on image and video datasets (see e.g. Higgins et al. (2017), Bouchacourt et al. (2017), or Goyal et al. (2017)), the application of VAEs to text has proven challenging. Specifically, Bowman et al. (2015) found that a VAE with an LSTM-based decoder fails to learn a useful latent code when trained naively – the approximate posterior collapses to the prior, $q_\theta(z|x) = p(z)$, for all inputs $x$, leading to a near independent relationship between $x$ and $z$.

A number of works have addressed this problem, mostly treating it as an optimization issue (Sønderby et al., 2016; Kingma et al., 2016; Yeung et al., 2017). However, as Chen et al. (2017)

point out, even if one could optimize exactly, the model would still learn trivial latent codes when using a high-capacity decoder such as an LSTM.

The reason for this is the VAE objective. Note that the log-likelihood (3) is only a function of the marginal distribution $p_\theta(x)$, whereas representation is an aspect of the joint distribution $p_\theta(x, z)$. That is, we approximately optimize a marginal quantity in the hope of producing the desired effect on the joint distribution. This approach is unreliable, although it works when the marginal imposes strong constraints on the joint distribution, such as when the decoder has limited capacity and the model is forced to use the latent structure to reach a high likelihood. However, in the high-capacity regime, e.g. when $p_\theta(x|z) = q(x)$ is close to achievable, the task of density estimation becomes disconnected from the goal of representation learning (Huszár, 2017).

We propose an alternative optimization problem that better captures the representation objective. Our idea is to explicitly control the amount of information stored in the latent code, as measured by the mutual information. In the next sections, we derive our procedure in detail and show that it enables representation learning with powerful decoders.

## 4 CONTROLLING INFORMATION IN LATENT CODE REPRESENTATIONS

Our goal is to learn a deep latent variable model $p_\theta(z, x) = p(z)\, p_\theta(x|z)$, while precisely controlling the coupling between the latent code $z$ and the output $x$. We formalize our goal in the following constrained optimization problem,

$$\max_\theta \quad \mathbb{E}_{x \sim q(x)} \left[ \log \int p_\theta(x|z)\, p(z)\, \mathrm{d}z \right], \tag{4}$$

$$\text{s.t.} \quad I_{p_\theta}(x, z) = M, \tag{5}$$

where $M \geq 0$ is a scalar constant denoting the desired *mutual information*, $I_{p_\theta}(x, z) = \mathbb{E}_{(x,z) \sim p_\theta}[\log \frac{p_\theta(x,z)}{p_\theta(x)\, p_\theta(z)}]$, between $x$ and $z$ as encoded in the model $p_\theta(z, x)$. When $M$ is close to zero, the code $z$ is uninformative about $x$, whereas a large value of $M$ approaching the entropy as maximum possible value, $H_{p_\theta}(x)$, should lead to a deterministic relation between $x$ and $z$. Intermediate values of $M$ will lead to lossy codes $z$ which recover a compressed representation of structure in $q(x)$ that can be most efficiently captured by $p_\theta(x|z)$.

## 5 THE MUTUAL AUTOENCODER (MAE)

We now describe our idea for approximately solving the problem (4–5). We call our approach the *mutual autoencoder*.

Our method relies on two results from the literature: 1) exact penalty functions (Zangwill, 1967) to accommodate the equality constraint (5) into the estimation problem; 2) the variational infomax bound, (Barber & Agakov, 2003), to approximate the intractable mutual information $I_{p_\theta}(x, z)$.

We briefly describe each of them and their role in the mutual autoencoder.

### 5.1 EXACT PENALTY FUNCTIONS

To approximately solve (4–5), we resort to a classic method for constrained nonlinear optimization, the *penalty function method*. First proposed by Zangwill (1967), this method uses a positive penalty constant $C > 0$ to relax (4–5) to the unconstrained problem

$$\max_\theta \quad \mathbb{E}_{x \sim q(x)} \left[ \log \int p_\theta(x|z)\, p(z)\, \mathrm{d}z \right] - C\, |I_{p_\theta}(x, z) - M|. \tag{6}$$

In (6), any deviation of $I_{p_\theta}(x, z)$ from $M$ is penalized linearly. For values of $C$ large enough, an unconstrained maximum of (6) recovers a feasible local maximum of (4–5). In particular, we do not need to take $C$ all the way to infinity, but a finite value is sufficient as shown by Han & Mangasarian (1979), and the magnitude of this value is determined by the unknown optimal Lagrange multiplier for (5), see Bertsekas (1999, Proposition 4.3.1). In practice, we observe that values of $C$ in the range $[0.1, 10]$ work best.

We will show experimentally that the above penalty approach is highly effective. However, to leverage the approach, we also need to approximate the intractable mutual information $I_{p_\theta}(x, z)$.

## 5.2 Variational Information Maximization (Infomax Inequality)

The mutual information $I_{p_\theta}(x, z)$ is intractable because of the complex structure in the conditional distribution $p_\theta(x|z)$. To overcome this difficulty, we leverage the variational infomax inequality of Barber & Agakov (2003).

**Theorem 1** (Variational infomax bound, Barber & Agakov (2003)). *For any two random variables $x$ and $z$ distributed according to the joint distribution with a density $p(x, z)$ and for any conditional density function $r(z|x)$ we have*

$$I_p(x, z) \geq H(z) + \mathop{\mathbb{E}}_{(x,z)\sim p}[\log r(z|x)]. \tag{7}$$

*Proof.* See Appendix B on page 12. $\qquad\square$

Equality in (7) is attained for $r(z|x) = p(z|x)$, so we can write

$$I_p(x, z) = H_p(z) + \max_r \mathop{\mathbb{E}}_{(x,z)\sim p}[\log r(z|x)]. \tag{8}$$

In our method, we constrain the inner search over $r$ to a parametric class, yielding the following lower bound approximation $\hat{I}_p(x, z) \leq I_p(x, z)$, to the mutual information:

$$\hat{I}_p(x, z) = H_p(z) + \max_\omega \mathop{\mathbb{E}}_{(x,z)\sim p}[\log r_\omega(z|x)]. \tag{9}$$

## 5.3 The Mutual Autoencoder

We are now in the position to combine the variational infomax bound (9) with the variational likelihood bound (ELBO) to define the following *mutual autoencoder objective* $L_m(p, q)$ to be maximized over the distributions $p$ and $q$.

$$L_m(p, q) := L(p, q) - C\left|\hat{I}_p(x, z) - M\right| \tag{10}$$

$$
\begin{aligned}
= \mathop{\mathbb{E}}_{x\sim q(x)}&\left[\mathop{\mathbb{E}}_{z\sim q(z|x)}[\log p(x|z)] - D_{\mathrm{KL}}(q(z|x)\,\|\,p(z))\right] \\
&- C\left|H_p(z) + \max_\omega \mathop{\mathbb{E}}_{(x,z)\sim p}[\log r_\omega(z|x)] - M\right|.
\end{aligned}
\tag{11}
$$

We now discuss how to realize this objective via gradient-based methods. The quantity $H_p(z)$ is typically a constant because we fix the prior $p(z)$ to a simple distribution such as a multivariate normal distribution. We assume the latent variable is continuous, that is $z \in \mathcal{Z}$ with $\mathcal{Z} = \mathbb{R}^k$, so we can use the reparametrization trick (Kingma & Welling, 2013; Rezende et al., 2014) to compute an unbiased and low-variance estimate of the gradients of the expectation over $q(z|x)$.

The difficult part is the derivative of the term $\max_\omega \mathbb{E}_{(x,z)\sim p}[\log r_\omega(z|x)]$ with respect to $p$. Denoting by $r_p^*$ the optimal $r_\omega$ corresponding to $p$, one can derive by *REINFORCE*-style reasoning (Williams, 1992) that

$$\nabla_p \max_\omega \mathop{\mathbb{E}}_{(x,z)\sim p}[\log r_\omega(z|x)] = \nabla_p \mathop{\mathbb{E}}_{(x,z)\sim p}[\log r_p^*(z|x)] \tag{12}$$

$$= \mathop{\mathbb{E}}_{(x,z\sim p)}\left[(\nabla_p \log p(z,x)) \cdot \log r_p^*(z|x) + \nabla_p \log r_p^*(z|x)\right]. \tag{13}$$

There are two difficulties here: 1) The last term in (13) that arises from the dependence of $r_p^*$ on $p$ is difficult to compute. 2) Evaluating $r_p^*$ requires solving a separate optimization program in $\omega$, which is too expensive to do for each gradient evaluation.

To address the first issue, we note that in the non-parametric limit, as $r_p^*(z|x) \approx p(z|x)$, the problematic term vanishes. We therefore ignore it, which is equivalent to treating $r_p^*$ as independent of $p$ during back-propagation. For the second issue, we keep a running estimate of the optimal $r_p^*$ and perform a single gradient update to it whenever $p$ is updated. This gives rise to the practical procedure described in Algorithm 1. A more detailed derivation of the method can be found in Appendix A.1.

---

**Algorithm 1** Mutual Autoencoder Training

---

1: **procedure** TRAINMAE$(\theta, \omega, B, C, M, N)$
2:     **for** $i = 1, \ldots, N$ **do**
3:         UPDATEMODEL$(\theta, \omega, B, C, M)$             // We simultaneously optimize the model...
4:         UPDATEMIESTIMATE$(\omega, \theta, B)$            // ...and the mutual information estimate.
5:     **end for**
6: **end procedure**

7: **procedure** UPDATEMIESTIMATE$(\omega, \theta, B)$
8:     Sample $(z_i, x_i) \sim p_\theta$ for $i = 1, \ldots, B$
9:     $g \leftarrow \frac{1}{B} \sum_{i=1}^{B} \nabla_\omega \log r_\omega(z_i|x_i)$           // Gradient estimate of the infomax bound.
10:     $\omega \leftarrow$ Update$(\omega, g)$
11: **end procedure**

12: **procedure** UPDATEMODEL$(\theta, \omega, B, C, M)$
13:     $g_{\text{ELBO}} \leftarrow$ EstimateElboGradient$(\theta)$
14:     $g_{\text{MI}} \leftarrow$ Estimate of $\nabla_\theta \mathbb{E}_{(x,z) \sim p_\theta}[\log r_\omega(z|x)]$    // Using reparametrization trick or *REINFORCE*.
15:     Sample $(z_i, x_i) \sim p_\theta$ for $i = 1, \ldots, B$
16:     $m \leftarrow H_p(z) + \frac{1}{B} \sum_{i=1}^{B} \log r_\omega(z_i|x_i)$        // Mutual information estimate.
17:     $\theta \leftarrow$ Update$(\theta, g_{\text{ELBO}} - C \cdot \text{sign}(m - M) \cdot g_{\text{MI}})$
18: **end procedure**

---

Note that unlike (13), the gradient of the term $\mathbb{E}_{(x,z) \sim p_\theta}[\log r_\omega(z|x)]$ with respect to $\theta$ can be efficiently approximated via the reparametrization trick (if $x$ is continuous) or *REINFORCE* (if $x$ is discrete). We provide a derivation of the necessary gradients in Appendix A.2.

## 6   DISCRETE DATA REQUIRES FLEXIBLE ENCODER DISTRIBUTIONS

Before presenting experimental results, we discuss the choice of encoder distribution, which we find to be critical to making our method work. More generally, it turns out that flexible encoder distributions are essential for VAE representation learning of discrete data. Below we give the formal statement and discuss its practical implications.

**Theorem 2.** *Consider a VAE model applied to discrete data. Assume that both the prior $p(z)$ and the approximate posterior $q(z|x)$ are Gaussian and that the decoder $p(x|z)$ is powerful enough to model the true marginal $q(x)$. Then the independent configuration given by $q(z|x) = p(z)$ and $p(x|z) = q(x)$ is* the only *global optimum of the VAE objective.*

*Proof.* Maximizing the VAE objective $L(p, q)$ is equivalent to minimizing the Kullback-Leibler divergence between the joint distributions $q$ and $p$, $D_{\text{KL}}(q(x)\,q(z|x)\,\|\,p(z)\,p(x|z))$. The VAE optimum is attained when the KL divergence is zero i.e. when $q(x, z) = p(x, z)$ almost everywhere. The independent solution clearly satisfies this.

To show that all other configurations are suboptimal, let $p, q$ be such that $q(z|x) \neq p(z)$ for some value of $x$ with $q(x) > 0$. Then the implied marginal $q(z) = \sum_{x \in \mathcal{X}} q(z|x)\,q(x)$ is a *finite* Gaussian mixture with at least one nonstandard component, so it cannot equal $p(z)$ and the KL divergence is positive. $\qquad\square$

Our theorem shows that not only is the VAE agnostic to representation learning, it in fact encourages the independent solution. The Gaussianity assumption introduces a gap due to $q(z)$ not being able to fit $p(z)$. We have observed this to be a major practical concern. The gap tends to grow with the dimension of the latent space as well as the amount of encoded information. We include 1D plots illustrating the issue in Appendix C.

In our experiments, we mitigate the issue by employing a resampling-based approximate posterior of Cremer et al. (2017) based on importance weighted autoencoders (Burda et al., 2015).

## 7 EXPERIMENTS

We show that the mutual autoencoder can learn latent codes ranging from independent to nearly deterministic. First, we consider two toy examples (a continuous and a discrete one), where we can visualize important quantities. Then we show promising results on text data, building a VAE model of movie reviews.

### 7.1 SPLITTING THE NORMAL

We compare our method to the variational autoencoder on the task of modeling the one-dimensional standard normal distribution, $q(x) = \mathcal{N}(x \mid 0, 1)$. Our goal here is to a) show in a minimal setting the issue of *representation collapse* in VAEs, and b) show that our method overcomes this problem and in fact can encode any pre-specified amount of information.

We now describe the experimental setup. The prior $p(z)$ is assumed to be a one-dimensional standard normal. Both the decoder $p_\theta(x \mid z)$ and encoder $q_\theta(z \mid x)$ are modeled as normal, with means and log-variances parametrized by a three-layer fully connected network. We train a VAE and several instances of the MAE with different values of the mutual information target $M$.

One can think of the task as splitting the normal $q(x)$ into an infinite mixture of normals $\int p(x \mid z) p(z) dz$. There exists a trivial optimum of the VAE objective $p_\theta(x \mid z) = q(z \mid x) = \mathcal{N}(x \mid 0, 1)$ that ignores the latent code. Indeed, this is the solution recovered by the VAE and the MAE with $M = 0$, corresponding to the top row of Figure 2a.

When we set $M > 0$, though, our method learns a non-trivial representation, as can be seen in Figure 2a. As we increase $M$, the conditionals $p_\theta(x \mid z)$ become more peaked and carry more information about $z$.

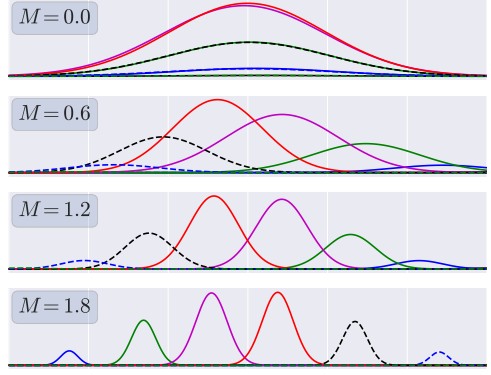
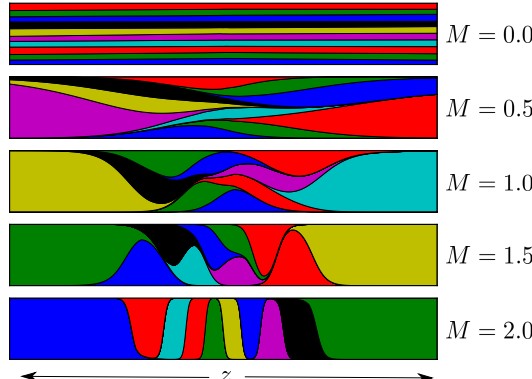

(a) MAE splits the normal. Each row corresponds to a MAE model with the specified mutual information target $M$. Each curve corresponds to the scaled conditional $p_\theta(x \mid z) p(z)$ for some fixed $z \in \{-2.1, -1.3, -0.4, +0.4, +1.3, +2.1\}$.

(b) Categorical example, modeling the uniform distribution over 10 labels. Each row corresponds to a MAE model with the specified mutual information target $M$. Each colored patch corresponds to one possible outcome of $x$. A vertical slice at position $z$ consists of 10 line segments whose lengths (sometimes 0) indicate the probabilities $p_\theta(x \mid z)$.

Figure 2: The mutual autoencoder learns codes ranging from independent to deterministic.

### 7.2 CATEGORICAL EXAMPLE

VAEs are particularly prone to ignoring the latent code when modeling discrete data. Here we show that the MAE is an effective solution even in this setting, and is able to learn a wide range of codes.

We consider synthetic data drawn from $q(x) = \text{Unif}[10]$. We let the latent variable again be a one-dimensional standard normal. The decoder is a three-layer fully connected network with a 10-way softmax output; the encoder is given by 10 separate normal distributions, one for each possible value of $x$.

As in the continuous case, the decoder $p_\theta(x|z)$ can easily reach an optimum of the VAE objective by outputting the marginal, Unif$[10]$, irrespective of $z$. What makes the discrete case more difficult is that *any* distribution over $\mathcal{X}$ can be represented independently of the latent $z$. Such a scenario is shown in the top row of Figure 2b; this is the model learned by the VAE and the MAE with $M = 0$.

For higher values of $M$, the MAE avoids this problem. In Figure 2b, we see that as $M$ increases, the conditionals $p_\theta(x|z)$ learned by MAE move from independence towards determinism.

## 7.3 MOVIE REVIEWS

We now demonstrate the effectiveness of the mutual autoencoder on real text data.

We consider a sentence modeling task using a subset of the IMDB movie review dataset (Diao et al., 2014). We split each review into sentences and extract those of length 8 words or shorter. We train several instances of the MAE with different $M$, using a variant of the bidirectional LSTM (Schuster & Paliwal, 1997; Hochreiter & Schmidhuber, 1997) as the encoder and a standard LSTM as the decoder. The conditioned value of the latent variable $z$ is fed to the decoder LSTM at each step.

To evaluate the information content of a given model's latent code, we perform a simple reconstruction experiment: we use the model to encode a random subset of the training data and decode the obtained representations back into sentence space. A model with a highly informative latent code should be able to approximately reconstruct the input. We consider two metrics: 1) the number of exactly reconstructed sentences, and 2) the number of matching words between the input sentence and the reconstruction.

Figure 3 shows the results for MAE models trained with different $M$. The graph shows a simple monotonic relation: as $M$ increases, so does the amount of encoded information, which in turn is reflected by the model's ability to autoencode. The MAE provides a powerful way of controlling this behavior by setting $M$.

In Table 1 we show sample sentence reconstructions of a VAE and two MAE models. As expected, the VAE fails to learn a useful latent code, $q_\theta^{\text{VAE}}(z|x) \approx p(z)$, so its 'reconstructions' are random samples from the decoder model. At the other extreme, the MAE with a high value of the target mutual information (last row) learns a close-to-deterministic code and reconstructs the input sentence with high fidelity, at the expense of sample diversity. MAEs trained with intermediate values of $M$ learn to 'paraphrase' input sentences.

To further inspect the learned representations, we interpolate between sentences in the latent space. An example is shown in Table 2. We see that the model picks up on syntactic elements such as specific word choice or sentence length and that the generated sentences are mostly grammatical (subject to limitations of the training data).

| Input | there are many great scenes of course . |
|---|---|
| VAE | and he knows it too . |
| | terrific performances from all three stars . |
| | this movie could have been a classic . |
| MAE ($M = 5$) | there are things that i liked . |
| | as a whole it works pretty well . |
| | there were a few good performances too . |
| MAE ($M = 10$) | there were many scenes like that . |
| | there are many great scenes of course . |
| | there were many scenes like that . |

Table 1: Sentence reconstructions. The input sentence is encoded and decoded by a given model, and the output is displayed. We show three reconstructions per model.

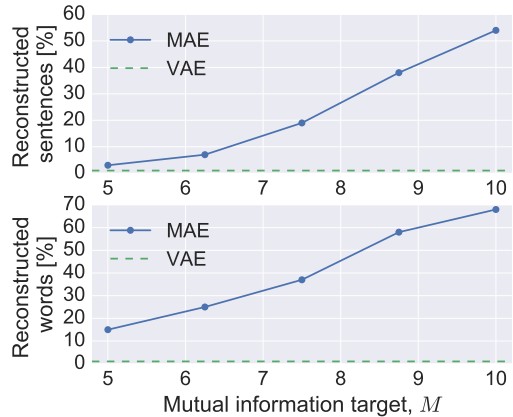

| |
|---|
| there are many great scenes of course . |
| there are other good people as well . |
| they so look real its incredible . |
| they like different , not good . |
| they like different , not good . |
| these people talk way too much . |
| die hard gets a few character . |
| i love these kind of movies . |
| i left out a character . |
| i love those films ) . |

Figure 3: Reconstruction ability grows with $M$.    Table 2: Sentence interpolation, $M = 10$.

## 8    RELATED WORK

To our knowledge, the first work to notice the failures of VAE-type models when combined with high-capacity decoders was (Bowman et al., 2015). Since then, a number of authors have reported similar difficulties and many fixes have been proposed.

Probably the most popular of these is 'KL cost annealing' (Bowman et al., 2015) or 'warm-up' (Sønderby et al., 2016), whereby the KL term in the VAE objective gets a multiplicative weight that gradually increases from zero to one during training. Another class of approaches is based on carefully limiting the capacity of the generative model class, as in (Chen et al., 2017), (Yang et al., 2017), or (Yeung et al., 2017). Finally, some works make use of auxiliary objectives: Kingma et al. (2016) introduce a 'free bits' constraint on the minimum value of the KL term, whereas (Semeniuta et al., 2017) add a reconstruction term of an intermediate network layer. Zhao et al. (2017) propose a more drastic change to the objective, either replacing the KL term by one derived from a different divergence or removing it completely.

Our approach has close connections to the work of Chen et al. (2016), who also use the variational infomax bound to learn meaningful representations. Their work is based on the GAN framework of Goodfellow et al. (2014) and hence may be difficult to apply to text. The work of Hu et al. (2017) can also be seen as employing (an approximation of) mutual information to enforce coupling between the latent and output variable, although they do not state it explicitly.

## 9    CONCLUSION

In this work, we address the problem of representation collapse in VAEs with high-capacity decoders. We argue that the VAE objective is not sufficient to encourage learning of useful representations and propose to explicitly control the amount of information stored in the latent code. We present new techniques for solving the resulting constrained optimization problem. We show experimentally that our method is highly effective at encoding the user-specified number of bits, and can thus learn a continuum of representation spaces with varying properties.

Our method has the advantage of being principled, preserving the probabilistic interpretation of VAEs. Also, it is independent of specific architectural choices; it can be combined with any decoder distribution and benefits from its capacity. We find optimization to be easy. On the downside, the method is considerably slower to train due to the extra mutual information term.

In future work, we aim to extend the method to the semi-supervised setting. Making use of a supervisory signal, we would be able to control not only the amount, but also the kind of information stored in the latent code. This would be an important step towards fully controllable representation learning.

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

## A  *REINFORCE* DERIVATIONS FOR SECTION 5.3

### A.1  DERIVATION AND PROPERTIES OF (13)

Note that we treat $p(z)$ as constant and $\nabla_p$ denotes the gradient with respect to $p(x|z)$ only.

$$\nabla_p \mathop{\mathbb{E}}_{(x,z)\sim p}[\log r_p^*(z|x)] = \nabla_p \mathop{\mathbb{E}}_{z\sim p}\left[\sum_{x\in\mathcal{X}} p(x|z)\cdot \log r_p^*(z|x)\right] \tag{14}$$

$$= \mathop{\mathbb{E}}_{z\sim p}\left[\sum_{x\in\mathcal{X}} (\nabla_p p(x|z))\cdot \log r_p^*(z|x) + p(x|z)\cdot\left(\nabla_p \log r_p^*(z|x)\right)\right] \tag{15}$$

Taking out $p(x|z)$ and writing the resulting ratio as the derivative of the log gives

$$= \mathop{\mathbb{E}}_{z\sim p}\left[\sum_{x\in\mathcal{X}} p(x|z)\big\{(\nabla_p \log p(x|z))\cdot \log r_p^*(z|x) + \nabla_p \log r_p^*(z|x)\big\}\right] \tag{16}$$

$$= \mathop{\mathbb{E}}_{z\sim p}\left[\sum_{x\in\mathcal{X}} p(x|z)\big\{(\nabla_p \log p(z,x))\cdot \log r_p^*(z|x) + \nabla_p \log r_p^*(z|x)\big\}\right] \tag{17}$$

$$= \mathop{\mathbb{E}}_{(x,z\sim p)}\left[(\nabla_p \log p(z,x))\cdot \log r_p^*(z|x) + \nabla_p \log r_p^*(z|x)\right]. \tag{18}$$

Next, we show that the term $\mathbb{E}_{(x,z\sim p)}\left[\nabla_p \log r_p^*(z|x)\right]$ vanishes when $r_p^*(z|x) = p(z|x)$.

$$\mathop{\mathbb{E}}_{(x,z\sim p)}\left[\nabla_p \log p(z|x)\right] = \mathop{\mathbb{E}}_{x\sim p}\mathop{\mathbb{E}}_{z\sim p(z|x)}\left[\frac{\nabla_p p(z|x)}{p(z|x)}\right] \tag{19}$$

$$= \mathop{\mathbb{E}}_{x\sim p}\left[\int_{\mathcal{Z}} \frac{\nabla_p p(z|x)}{p(z|x)}\cdot p(z|x)\mathrm{d}z\right] = \mathop{\mathbb{E}}_{x\sim p}\left[\nabla_p \int_{\mathcal{Z}} p(z|x)\mathrm{d}z\right] = 0. \tag{20}$$

### A.2  GRADIENT OF THE MAE OBJECTIVE

Denote

$$\hat{I}(\theta) := H_p(z) + \mathop{\mathbb{E}}_{(x,z)\sim p_\theta}[\log r_\omega(z|x)]. \tag{21}$$

We will compute the gradient of $C\left|\hat{I}(\theta) - M\right|$ with respect to the model parameters $\theta$.

$$\nabla_\theta C\left|\hat{I}(\theta) - M\right| = C\cdot \mathrm{sign}\left(\hat{I}(\theta) - M\right)\cdot \nabla_\theta \hat{I}(\theta) \tag{22}$$

$$= C\cdot \mathrm{sign}\left(\hat{I}(\theta) - M\right)\cdot \nabla_\theta \mathop{\mathbb{E}}_{z\sim p}\left[\mathop{\mathbb{E}}_{x\sim p_\theta(x|z)}[\log r_\omega(z|x)]\right] \tag{23}$$

$$= C\cdot \mathrm{sign}\left(\hat{I}(\theta) - M\right)\cdot \mathop{\mathbb{E}}_{z\sim p}\left[\sum_{x\in\mathcal{X}} (\nabla_\theta p_\theta(x|z))\cdot \log r_\omega(z|x)\right] \tag{24}$$

Again making use of the derivative of the log, we get

$$= C\cdot \mathrm{sign}\left(\hat{I}(\theta) - M\right)\cdot \mathop{\mathbb{E}}_{z\sim p}\left[\sum_{x\in\mathcal{X}} p_\theta(x|z)(\nabla_\theta \log p_\theta(x|z))\cdot \log r_\omega(z|x)\right] \tag{25}$$

$$= C\cdot \mathrm{sign}\left(\hat{I}(\theta) - M\right)\cdot \mathop{\mathbb{E}}_{(x,z)\sim p_\theta}\left[(\nabla_\theta \log p_\theta(x|z))\cdot \log r_\omega(z|x)\right]. \tag{26}$$

The gradient in (26) can be estimated using Monte Carlo sampling of both the expectation and the $\hat{I}(\theta)$ term inside the sign. The sign estimate will inevitably be wrong sometimes, making our gradient estimator biased. However, note that only the norm is affected, while the direction of the gradient estimate is still correct in expectation.

## B INFOMAX INEQUALITY

The inequality was proven by Barber & Agakov (2003). Here we provide an alternative direct proof.

*Proof.* The following Gibbs inequality yields

$$D_{\text{KL}}(p(x,z) \,\|\, r(z|x)\, p(x)) \geq 0 \tag{27}$$

$$\Leftrightarrow \mathop{\mathbb{E}}_{(x,z)\sim p} [\log p(z|x) + \log p(x) - \log r(z|x) - \log p(x)] \geq 0 \tag{28}$$

$$\Leftrightarrow \mathop{\mathbb{E}}_{(x,z)\sim p} [\log p(z|x)] \geq \mathop{\mathbb{E}}_{(x,z)\sim p} [\log r(z|x)]. \tag{29}$$

By the definition of mutual information and conditional entropy, we have

$$I_p(x,z) = H_p(z) - H_p(z|x) \tag{30}$$

$$= H_p(z) + \mathop{\mathbb{E}}_{(x,z)\sim p} [\log p(z|x)] \tag{31}$$

$$\geq H_p(z) + \mathop{\mathbb{E}}_{(x,z)\sim p} [\log r(z|x)]. \tag{32}$$

$\square$

## C POSTERIOR APPROXIMATION

We found encoder flexibility to be instrumental to training a good MAE model. Here we illustrate the effect on a toy task of modeling $x \sim \text{Unif}[10]$ with a one-dimensional normal latent space. We set the mutual information target $M$ to the theoretical maximum, aiming to learn a deterministic code.

Figure 4 shows the difference between using the standard Gaussian encoder and a more complex resampling-based encoding distribution of Cremer et al. (2017) with $k = 5$ samples. We see that not only does the latter learn better approximate posteriors (bottom row), it also enables the decoder to train more effectively, as reflected in a sharper, closer-to-deterministic model posteriors (top row).

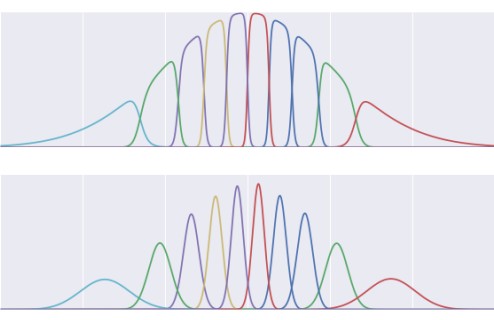
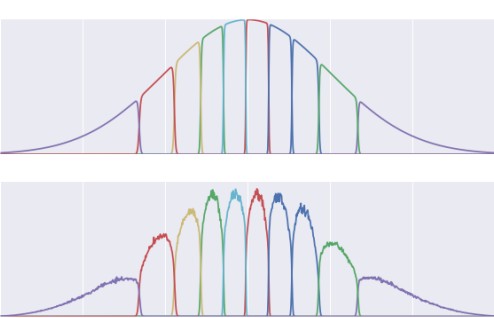

(a) MAE trained with a Gaussian encoder. Top: true model posterior $p_\theta(z|x)$. Bottom: posterior approximation $q_\theta(z|x)$. Each curve corresponds to one possible value of $x$.

(b) MAE trained with a resampling-based encoder. Top: true model posterior $p_\theta(z|x)$. Bottom: posterior approximation $q_\theta(z|x)$. Each curve corresponds to one possible value of $x$.

Figure 4: The effect of encoder flexibility.

