# OpenReview forum: "The Mutual Autoencoder: Controlling Information in Latent Code Representations"
_ICLR.cc/2018/Conference — Reject_

### Official Review · AnonReviewer2 · 2017-11-16
**Review: Interesting use of a mutual information target, but not sure if it's actually a target.**

**Rating:** 4
**Confidence:** 5

**Review:**


Summary

This paper proposes a penalized VAE training objection for the purpose of increasing the information between the data x and latent code z.  Ideally, optimization would consist of maximizing log p(x) - | I(x,z) - M |, where M is the user-specified target mutual information (MI) and I(x,z) is the model’s current MI value, but I(x,z) is intractable, necessitating the use of an auxiliary model r(z|x).  Optimization, then, consists of alternating gradient ascent on the VAE parameters and r’s parameters.  Experiments on simulations and text data are reported, showing that increasing M has the desired effect of allowing more deviation from the prior.  Specifically, this is shown through text generation where the sampled sentences become more varied as M is decreased and better reconstructed as M is increased.


Evaluation

Pros:  I like how this paper formalizes failure in representation learning as information loss in z---although the formulation is not particularly novel, i.e. [Zhao et al., ArXiv 2017]), and constructs an explicit, penalized objective to allow the user to specify the amount of information retained in z.  In my opinion, the proposed objective is more transparent than the objectives proposed by related work.  For instance, Chen et al.’s (2017) Lossy VAE, while aiming to solve essentially the same problem, does so by parameterizing the prior and using a windowed decoder, but there is no explicit control mechanism as far as I’m aware (except for how many parameters / window size).  Perhaps the Beta-VAE’s [Higgins et al., ICLR 2017] KLD weight is similarly interpretable (as beta increases, less information is retained), but I like that M has the clear meaning of mutual information---whereas the beta in the Beta-VAE is just a Lagrangian.  In terms of experiments, I like the first simulation; it’s a convincing sanity check.  As for the second, I like the spirit of it, but I have some criticisms, as I’ll explain below.

Cons:  The method requires training an auxiliary model r(z|x) to estimate I(x,z).  While I don’t find the introduction of r(z|x) problematic, I do wish there was more discussion and analysis of how well the mutual information is being approximated during training, especially given some of the simplifying assumptions, such as r(z|x)=p(z|x).  If the MI estimate is way off, that detracts from the method and makes an alternative like the Beta-VAE---which doesn’t require an auxiliary model---more appealing, since what makes the MAE superior---its principled targeting of MI---does not hold in practice.

As for the movie review experiment, I find the sentence samples a bit anecdotal.  Was the seed sentence (“there are many great scenes of course”) randomly chosen or hand picked?  Was this interpolation behavior typical?  I ask these questions because I find the plot in Figure 3 all but meaningless.  It’s good that we see reconstruction quality go up as M increases, as expected, but the baseline VAE model is a strawman.  How does reconstruction percentage look for the Bowman et al. (2015) VAE?  What about the Beta-VAE?  Or Lossy VAE?  Figure 3 would be okay if there were more experiments, but as it is the only quantitative result, more work should have gone in to it.  For instance, a compelling result would be if we see one or more of the models above plateau in reconstruction percentage and the MAE surpass that plateau.


Conclusions

While I found aspects of this paper interesting, I recommend rejection primarily for two reasons.  The first is that I would like to see how well the mutual information is being estimated during training.  If the estimate is way off, this makes the method less appealing as what I like about it---the interpretable MI target---is not really a ‘target’, in practice, and rather, is a rough hyperparameter similar to the Beta-VAE’s beta term (which has the added benefit of no auxiliary model).  The second reason is the paper’s weak experimental section.  The only quantitative result is Figure 3, and while it shows reconstruction percentage increases with M, there is no way to contextualize the number as the only comparison model is a weak VAE, which gives ~ 0%.  Questions I would like to see answered: How good is the MI estimate?  How close is the converged VAE to the target?  How does the model compare to the Bowman et al. VAE or the Beta-VAE?  (It would be quite compelling to show similar or improved performance without the training tricks used by Bowman et al.)  Can we somehow estimate the appropriate M directly from data (such as based on the entropy of training or validation set) in order to set the target rigorously?


1.  S. Zhao, J. Song, and S. Ermon.  “InfoVAE: Information Maximizing Variational Autoencoders.”  ArXiv 2017.
2.  X. Chen, D. Kingma, T. Salimans, Y. Duan, P. Dhariwal, J. Shulman, I. Sutskever, and P. Abbeel.  “Variational Lossy Autoencoder.”  ICLR 2017.
3.  I. Higgins, L. Matthey, A. Pal, C. Burgess, X. Glorot, M. Botvinick, S. Mohamed, and A. Lerchner. “Beta-VAE: Learning Basic Visual Concepts with a Constrained Variational Framework.”  ICLR 2017
4.  S. Bowman, L. Vilnis, O. Vinyas,  A. Dai, R. Jozefowicz, and S. Bengio.  “Generating Sentences from a Continuous Space.”  CoNLL 2016.

---

### Official Review · AnonReviewer1 · 2017-11-24
**An interesting paper, but needs more work**

**Rating:** 5
**Confidence:** 4

**Review:**

This paper presents mutual autoencoders (MAE). MAE aims to address the limitation of regular variational autoencoders (VAE) for latent representation learning — VAE sometimes simply ignores the latent code z, especially with a powerful decoding distribution. The idea of MAE is to optimize the VAE objective subject to a constraint on the mutual information between the data x and latent code z: setting the mutual information constraints larger will force the latent code z to learn a meaningful representation of the data. An approximation strategy is employed to approximate the intractable mutual information. Experimental results on both synthetic data and movie review data demonstrate the effectiveness of the MAEs.

Overall, the paper is well-written. The problem that VAEs fail to learn a meaningful representation is a well-known issue. This paper presents a simple, yet principled modification to the VAE objective to address this problem. I do, however, have two major concerns about the paper:

1. The proposed idea to add a mutual information constraint between the data x and latent code z is a very natural fix to the failure of regular VAEs. However, mutual information itself is not a quantity that is easy to comprehend and specify. This is not like, e.g., l2 regularization parameter, for which there exists a relatively clear way to specify and tune. For mutual information, at least it is not clear to me, how much mutual information is “enough” and I am pretty sure it is model/data-dependent. To make it worse, there exist no metrics in representation learning for us to easily tune this mutual information constraint. It seems the only way to select the mutual information constraint is to qualitative inspect the model fits. This makes the method less practical.

2. The approximation to the mutual information seems rather loose. If I understand correctly, the optimization of MAE is similar to that of a regular VAE, with an additional parametric model r_w(z|x) which is used to approximate the infomax bound. (And this also adds an additional term to the gradient wrt \theta). r_w(z|x) is updated at the same time as \theta, which means r_w(z|x) is quite far from being an optimal r* as it is intended, especially early during the optimization. Further more, all the derivation following Eq (12-13) are based on r* being optimal, while in reality, it is probably not even close. This makes the whole approximation quite hand-waving.

Related to 2, the discussion in Section 6 deserves more elaboration. It seems that having a flexible encoder is quite important, yet the authors only mention lightly that they use the approximate posterior from Cremer et al. (2017). Will MAE not work without this? How will VAE (without the mutual information constraint) work with this? A lot of the details seem to be glossed over.

Furthermore, this work is also related to the deep variational information bottleneck of Alemi et al. 2017 (especially in the appendix they derived the VAE objective using information bottleneck principle). My intuition is that using a larger mutual information constraint in MAE is somewhat similar to setting the regularization \beta to be smaller than 1 — both are making the approximating posterior more concentrated. I wonder if the authors have explored this idea.


Minor comments:

1. It would be more informative to include the running time in the presented results.

2. Since the goal of r_w(z | x) is to approximate the posterior p(z | x), what about directly using q(z | x) to approximate it?

3. In Algorithm 1, should line 14 and 15 be swapped? It seems samples are required in line 14 as well.

4. Nitpicking: technically the model in Eq (1) is not a hierarchical model.

---

### Official Review · AnonReviewer3 · 2017-11-29
**THE MUTUAL AUTOENCODER: CONTROLLING INFORMATION IN LATENT CODE REPRESENTATIONS**

**Rating:** 4
**Confidence:** 4

**Review:**

The authors propose a variational autoencoder constrained in such a way that the mutual information between the observed variables and their latent representation is constant and user specified. To do so, they leverage the penalty function method as a relaxation of the original problem, and a variational bound (infomax) to approximate the mutual information term in their objective.

I really enjoyed reading the paper, the proposed approach is well motivated and clearly described. However, the experiments section is very weak. Although I like the illustrative toy problem, in that it clearly highlights how the method works, the experiment on real data is not very convincing. Further, the authors do not consider a more rigorous benchmark including additional datasets and state-of-the-art modelling approaches for text.

- {\cal Z} in (1) not defined, same for \Theta.

---

### Author Response · Authors · 2018-01-02
**Thank you**

We would like to thank the reviewers for reading the paper so carefully and for their detailed reviews. The main weaknesses of the paper seem to be the experimental section and a missing analysis of how well MI is estimated by our method. We will definitely work on this and submit to a later conference.

---

### Decision · Program_Chairs · 2018-01-29
**ICLR 2018 Conference Acceptance Decision**

**Decision:**

Reject

**Comment:**

This is a well-written paper that aims to address an important problem. However, all the reviewers agreed that the experimental section is currently too weak for publication. They also made several good suggestions about improving the paper and the authors are encouraged to incorporate them before resubmitting.